# Wind tunnel study of yawed porous discs subjected to veered inflow

Shantanu Purohit<sup>a</sup>, Haoyuan Sun<sup>a</sup>, Andrea Sciacchitano<sup>a</sup>, and Wei Yu<sup>a</sup>

<sup>a</sup>Faculty of Aerospace Engineering, Delft University of Technology, Delft, the Netherlands

**Correspondence:** Shantanu Purohit (s.purohit@tudelft.nl)

Abstract. Atmospheric boundary layer flow during stably stratified conditions often exhibits wind veering—the change in wind direction with height—which significantly influences wind turbine wake dynamics and its downstream recovery. This study investigates the impact of veered inflow on turbine wakes through wind tunnel experiments using high-resolution stereo particle image velocimetry (SPIV). A porous disc of uniform porosity is employed as a surrogate for wind turbines to systematically examine wake characteristics under both non-yawed and yawed conditions. The results reveal that veered inflow induces an ellipsoidal-shaped wake for a non-yawed porous disc. Under yawed conditions, however, the interaction between yaw and veer leads to a complex wake shape, where the curled shape due to yaw is superimposed on the wake stretching due to veer. Furthermore, the strength of the two counter-rotating vortex pairs formed around yawed discs is reduced due to wind veering. A budget analysis of the streamwise momentum equation is performed to shed light on the mechanism of wake recovery. The results demonstrate that wind veering leads to faster wake recovery and more available power for downstream wind turbines. These findings imply that, under conditions of extreme wind veer, yawing the turbine may offer limited additional energy recovery, as wind veering alone facilitates significant wake re-energization.

#### 1 Introduction

Wind turbines (WTs) operate within the lowest levels of the atmospheric boundary layer, where atmospheric conditions vary significantly during the diurnal cycle. During the morning hours, when the sun warms the surface, buoyancy dominates shear in driving turbulence, leading to strong vertical mixing and convective updrafts (Wurps et al., 2020). In contrast, during nighttime, radiative cooling at the surface suppresses buoyancy, and turbulence is primarily generated by wind shear. The resulting stable boundary layer (SBL) is characterized by weak turbulence and limited vertical mixing, leading to stratification of the flow. A characteristic feature of the nighttime SBL is the Coriolis-force-induced wind veer, which refers to the change in wind direction with height. Wind veer tends to be more pronounced under stable stratification compared to unstable conditions due to the suppression of vertical mixing in stable layers (Churchfield and Sirnivas, 2018). As wind turbines continue to grow in size, the impact of wind veer on their performance becomes increasingly critical. The largest commercially deployed wind turbine to date—the SG 14-222 DD at the Moray West offshore wind farm in Scotland—features a rotor diameter of 260 meters and a power capacity of 14.7 MW (Siemens Gamesa, 2024). Prototypes of even higher-rated turbines, exceeding 20 MW, are currently in various stages of development, which will result in even taller structures. At such heights, wind veer can be substantial; for example, wind veering angles up to 40° have been observed at 200 m height at the Cabauw observatory

30

(Van Ulden and Holtslag, 1985). Furthermore, wind veer was observed to occur more than 70% of the time over the course of a year in offshore environments (Bodini et al., 2019), highlighting the need to account for its effect on WT wake structure and evolution.

Wind veer has been shown to significantly influence the wake characteristics and power performance of WTs in many numerical and field experiments. Using lidar and turbine data, Sanchez Gomez and Lundquist (2020) reported turbine underperformance at high wind veer, while analysis of a 5-year field dataset by Gao et al. (2021) also found a loss in power production of up to 6.5% in veering wind conditions. Large-eddy simulation (LES) is the most preferred way to study wind turbine wakes due to its ability to resolve large-scale turbulent structures and accurately capture unsteady wake dynamics (Xie and Archer, 2017). Several numerical studies have demonstrated a skewed wake profile as a result of veered inflow for non-yawed turbines (Lu and Porté-Agel, 2011; Abkar and Porté-Agel, 2016; Vollmer et al., 2016; Bromm et al., 2017; Xie and Archer, 2017; Churchfield and Sirnivas, 2018; Wu et al., 2024; Klemmer and Howland, 2024). Churchfield and Sirnivas (2018) hypothesized that under veer conditions, the wake adopts an ellipsoidal shape, whereby high-momentum flow can reach the wake core more quickly due to the shorter lateral distance along the minor axis of the ellipsoid, thereby enhancing wake recovery. Abkar and Porté-Agel (2016) attributed faster wake recovery under veered conditions to increased shear production and enhanced turbulent kinetic energy resulting from the combined effects of vertical and lateral shear, in contrast to that of a turbine operating under unidirectional inflow. The stretching of the turbine wake due to ambient wind veer was also observed in lidar measurements-based field studies (Bodini et al., 2017).

As wind turbine wake interactions are one of the leading causes of reduced power production (Barthelmie et al., 2009) and mechanical and fatigue loading on downstream wind turbines (Sathe et al., 2013), wake steering has emerged as an effective way to mitigate wake losses (Gebraad et al., 2016). The physics of wake steering and its benefits as a yaw control strategy have been extensively investigated in numerical simulations (Fleming et al., 2014; Vollmer et al., 2016; Archer and Vasel-Be-Hagh, 2019) and both laboratory (Bastankhah and Porté-Agel, 2016; Bartl et al., 2018; Schottler et al., 2018; Hulsman et al., 2022) and field experiments (Fleming et al., 2017, 2019, 2020; Howland et al., 2019, 2022). Usually, the effect of ambient wind veer on the efficacy of wake steering as a yaw control strategy is not considered. Several numerical studies have begun to explore yawed turbine wakes under veered inflow. For instance, Narasimhan et al. (2022) showed that wind veer distorted the structure of the two counter-rotating vortices and introduced an asymmetry in the curled shape of the wake. In a conventionally neutral boundary layer (CNBL), they found that the influence of veer could be superimposed on the yawinduced wake behavior. Including wind veer effects and turbine yaw in analytical wake models has started to receive traction recently. For instance, Mohammadi et al. (2022) extended the vortex-sheet curled wake model of Bastankhah et al. (2022) by incorporating veer effects through a height-dependent effective yaw angle term. The modified model showed good agreement with the LES results. More recently, Narasimhan et al. (2025) proposed a new analytical wake model for turbines (both yawed and unyawed) operating in CNBL and SBL conditions, also showing strong agreements with the LES data. However, the influence of indirect atmospheric forcing, such as wind veer, on yawed rotors has not been extensively investigated, despite its importance for understanding the efficacy of yaw-based wake steering. Wake steering via yaw misalignment is generally most effective under low-turbulence conditions (Fleming et al., 2019; Simley et al., 2020), which are typically found at night

65

80

85

and are associated with a stable boundary layer. Notably, wind veer is also prevalent in nighttime stable conditions, making it essential to investigate the combined effects of wind veer and yaw on wake behavior for more effective deployment of yaw control strategies in real-world scenarios.

Complementary to these modeling efforts, wind tunnel experiments offer valuable insights into the flow physics of yawed wind turbines under controlled inflow conditions. Numerous studies have explored yawed turbines, driven by the potential of yaw misalignment as a strategy for wake control. A brief review of experimental works on yawed wind turbines is presented here. The aerodynamics of yawed wind turbines were first studied experimentally by (Grant et al., 1997) and (Grant and Parkin, 2000). In the former study, the authors visualized the motion of vortex shedding for a wind turbine in yaw and found the power coefficient to be dependent on yaw angles, in alignment with the earlier theoretical works. Whereas, in the latter study, the authors used digital PIV to measure velocity fields and tip vorticity of a turbine in yaw, and found that the initial formation of the tip vortex depends on the rotor yaw angle and blade orientation. The tip vortex of the yawed wind turbine was investigated in the open-jet facility at TU Delft by (Haans et al., 2005). The authors found the expansion of the skewed wake to be strongly correlated with  $C_T$  and to play a key role in governing the wake skew angle. Later, an analytical expression of wake skew angle dependence on  $C_T$  and turbine yaw angle was given by (Jiménez et al., 2010). In a seminal experimental work by (Medici and Alfredsson, 2006), the authors first noted the asymmetric wake shape owing to turbine yaw as a result of the lateral force exerted by the turbine on the flow, as well as the importance of wake rotation on wake development. In a wind tunnel study of wake interference between two wind turbines, (Adaramola and Krogstad, 2011) recommended yawing the upstream turbine in order to increase the overall power production in wind farms.

The asymmetric and curled shape of the wake of yawed turbines was first observed by (Howland et al., 2016), where they used hot wire anemometry and Pitot-static probes for wake measurements of a porous disc. They ascribed the curled shape of the wake to the two counter-rotating vortices shed by the yawed porous disc. Concurrently, in the seminal work of (Bastankhah and Porté-Agel, 2016), the authors studied the wake of yawed WTs experimentally using SPIV and investigated the formation mechanism of counter-rotating vortex pairs that lead to the curled shape of the wake, which they attributed to the strong spanwise velocity in the wake. Moreover, they also proposed an analytical model for wake deflection and velocity distribution in the far wake. To investigate the impact of inflow turbulence and inflow shear on yawed wind turbines, (Bartl et al., 2018) used laser Doppler anemometry (LDA) to measure wake flow. Their results indicate the dependence of wake shape on turbulence level in the inflow, as the asymmetry in the yawed turbine wake reduces as a result of enhanced mixing for increased levels of inflow turbulence. However, as the inflow shear they considered was only moderate, not much impact of shear on the wake was observed. In another work, (Schottler et al., 2018) performed wind tunnel experiments on two different turbines under both non-yawed and yawed conditions and characterized wake width based on turbulence intermittency parameter and found a velocity increment region surrounding the mean deficit wake region and high TKE region, making the wake width considerably larger, which is important to consider in wake steering-based control strategies. More recently, (Hulsman et al., 2022) studied the effect of boundary layer and turbulence intensity on the curled shape of yawed wind turbines in a wind tunnel experiment. The authors found that boundary layer inflow accelerates the formation of the curled shape sooner compared to the uniform inflow case due to shear in the wind, wake rotation, and the formation of counter-rotating vortex pairs as a result of yaw.

https://doi.org/10.5194/wes-2025-185 Preprint. Discussion started: 21 October 2025

As highlighted in the preceding literature review, most wind tunnel studies of yawed wind turbines have employed idealized inflow conditions. Atmospheric stability induces two important indirect forcings: vertical wind shear and wind veer (Klemmer and Howland, 2024). While the impact of vertical wind shear on wind turbine wake characteristics is often studied in wind tunnel experiments, wind veer has not been experimentally investigated. Furthermore, understanding wake behavior under veered inflow is essential, as the effectiveness of wake redirection strategies, such as wake steering, depends on the interaction between yaw misalignment and realistic atmospheric conditions (Vollmer et al., 2016).

This study aims to experimentally assess the wake behavior of a statically yawed porous disc subjected to veered inflow. The choice of a non-rotating porous disc is motivated by the fact that they have been commonly employed as a surrogate for modeling wake characteristics of both horizontal-axis (Howland et al., 2016) and vertical-axis wind turbines (Huang et al., 2022). Previous studies have demonstrated that porous discs can reproduce the key features of turbine wake, particularly in the far wake. For instance, Aubrun et al. (2013) found that beyond x/D > 3, the wakes of a rotating turbine and a porous disc exhibit similar characteristics. Likewise, Lignarolo et al. (2016) reported comparable wake expansion and energy extraction between a wire mesh disc and a wind turbine under low turbulence when matched in diameter and thrust coefficient. Good agreement in far wake statistics has also been observed by Neunaber et al. (2021) and Vinnes et al. (2022), while similar wake characteristics for both models under varying turbulence levels were reported by Öztürk et al. (2023). Moreover, porous discs have also been effectively used in model wind farm studies (Camp and Cal, 2016).

To the best of the authors' knowledge, this is the first-of-its-kind study that attempts to study the impact of wind veer on wind turbines in a laboratory setting. In the current experiment, the wind turbine is represented as a non-rotating porous disc, and stereoscopic particle image velocimetry (SPIV) is used as a flow measurement technique. The current wind tunnel experiments for yawed discs and veered inflow in controlled conditions can provide validation data for high-fidelity numerical modeling codes. Moreover, this work can serve as a foundation for improving engineering wake models for yawed turbines proposed in many studies in the past to incorporate veer effects. Apart from investigating the feasibility of studying wind veer in wind tunnel experiments, we aim to answer the following research questions:

- 1. How do wind veer-induced wake stretching and yaw-induced wake curl enhance wake recovery?
- 2. What is the impact of wind veering on the effectiveness of wake steering-based control strategies?

The rest of the paper is organized as follows: In Section 2, the experimental setup and methodology are outlined, followed by discussions of results in Section 3. The implications of the results on wake steering and final conclusions are presented in Section 4.

## 125 2 Experimental method

This section outlines the experimental setup and methodology used to investigate the effects of veer on porous disc wakes. Section 2.1 introduces the schematic and description of the wind tunnel setup, followed by the design of the porous disc and wind veer model in Section 2.2. The test matrix and experimental conditions are summarized in Section 2.3, while details of the

SPIV measurement system are provided in Section 2.4. The associated flow measurement uncertainty is discussed in Section 2.5. Finally, Section 2.6 presents the flow characterization in the absence of the porous disc.

## 2.1 Wind tunnel

The experiments are performed in the W-tunnel at TU Delft Aerospace Engineering Laboratories. The W-tunnel is an openjet wind tunnel with an adjustable test section at the outlet. In this experiment, the test section at the exit has dimensions  $0.6~m \times 0.6~m$ , which is large enough for deflected flow as a result of yaw and wind veer to remain within the streamtube. The measurements are carried out at the free-stream velocity  $U_{\infty} = 10.3~ms^{-1}$ . At this flow velocity, the Reynolds number based on disc diameter is approximately 68,000, and the turbulence in the undisturbed flow is of the order of 0.5%. A rough schematic sketch in Fig. 1 shows the experimental setup employed in the current study to measure wake cross-section (Fig. 1a) and wake propagation in streamwise planes (Fig. 1b). Figure 2 shows a photograph of the experimental setup for cross-stream plane measurements, with key components labeled.

#### 2.2 Porous disc and wind veer model

A uniform porous disc of diameter 10 cm and porosity 0.6 was 3D-printed to replicate the effects of a wind turbine. Load measurements were performed to determine the thrust coefficient of the porous disc for the no-veer case and non-yawed conditions using a KD24S 2N (ME-MEßsysteme GmbH) force sensor with an accuracy of 0.1% of the full-scale value (2N). Based on the load measurements, the thrust coefficient ( $C_T$ ) of the porous disc is approximately 0.69, which also aligns well with the general trend of porosity vs.  $C_T$  curve given in the literature (Lignarolo et al., 2016; Huang et al., 2022).

The airfoil shape used in wind veering vanes is a NACA 0014 with a chord (c) of 0.2 m. This airfoil was selected based on its symmetric shape and relatively high angle of attack at which stall occurs (i.e.,  $\alpha \sim 14^{\circ}$ ). Two wind veering vanes are employed in this study: one producing a wind veer of  $10^{\circ}$  and the other a veer of  $20^{\circ}$  across the porous disc. The variation of angle of attack with height is shown in the left of Fig. 3. The vanes are designed so that at the center of the porous disc, the angle of attack is zero and increases symmetrically in positive and negative directions above and below the disc center. This is achieved by twisting the airfoil in opposite directions from the center outward, extending to the edges of the wind tunnel exit. For instance, in the  $10^{\circ}$  veer configuration, the vane twist varied linearly with a gradient of  $1^{\circ}$ /cm until the edges of the wind tunnel, resulting in a total wind veer of  $10^{\circ}$  across the porous disc ( $+5^{\circ}$  at one edge of the disc and  $-5^{\circ}$  at the opposite edge of the disc). This design approach reflects common assumptions in stable boundary layer simulations, where the wind veer angle is set to zero at hub height and varies in the opposite direction above and below the hub height.

However, such a continuous transition in angles from the disc center to the edges of the wind tunnel exit for the wind veer model of  $20^{\circ}$  would result in excessively high angles of attack, which are undesirable. To address this, the vane is twisted smoothly from  $0^{\circ}$  at the center of the disc to  $\pm 10^{\circ}$  at 5 cm away from the center. Beyond this point, the twist does not increase further—instead, the vane keeps a constant angle of  $\pm 10^{\circ}$  all the way out to the edges of the wind tunnel exit (see orange line in the left Fig. 3). This configuration ensures a smooth gradient of angle across the span of the guiding vane, as shown in Fig. 3 (right), where the wind veering model is placed at the exit of the wind tunnel.

Figure 1. Top view of the SPIV experimental setup for imaging (a) cross-stream yz-planes and (b) streamwise planes (xy). The cross-stream positions where measurements are taken are indicated by green dashed lines in Fig. 1a. In total, measurements were performed at four cross-stream planes. Measurements were also performed in the horizontal plane xy-plane, and the imaging locations are indicated by the solid red, green, and blue lines, with some overlap between consecutive planes. (c) The dimensions of the field of view (FOV) are indicated here. For streamwise FOVs, an overlap of 1D is applied at the center of the trapezoid. A Gaussian function is applied to stitch the streamwise FOVs. It should be noted that the dimensions of the FOV remain the same for both cross-stream and streamwise measurements.

#### 2.3 Case overview

The test matrix of the measured cases is summarized in Table 1. As it is of interest to investigate varying degrees of wind veer and its impact on wake recovery, two veered inflows are examined in addition to the baseline uniform inflow (no-veer) for comparison. Initially, flow characterization is conducted in the absence of the porous disc for all three inflows (uniform flow, 10° veer, and 20° veer). Subsequently, wake measurements are performed in the presence of the disc at four cross-stream

**Figure 2.** Photograph of the experimental setup in the W-Tunnel for measuring cross-stream wake planes. The key labeled components are as follows: 1. Exit of the W-tunnel; 2. Rotation stage on which a porous disc is mounted; 3. Wind veer model installed at the exit of the wind tunnel; 4. Laser sheet (span of the sheet marked by dashed green lines); 5. Traverse system; 6: Camera 1; 7: Camera 2. The field of view (FOV) is denoted in a filled green trapezoid. The coordinate system is shown on the bottom right.

**Table 1.** Test matrix used in the experiment

| Case                             | Turbine yaw angle (deg) | Wind veer across the disc (deg) |  |  |
|----------------------------------|-------------------------|---------------------------------|--|--|
| Flow characterization (w/o disc) | NA                      | 0, 10, 20                       |  |  |
| Uniform inflow                   | 0, 10, 20, 30           | 0                               |  |  |
| Veered inflow                    | 0, 10, 20, 30           | 10, 20                          |  |  |

planes and three streamwise planes, as illustrated in Fig. 1. The origin of the coordinate system is at the porous disc center, with x being the streamwise direction, and y and z denoting the spanwise and vertical directions, respectively.

# 2.4 Flow measurement system

Due to the combined effect of turbine yaw and wind veer, all three velocity components of the flow become relevant. Consequently, stereoscopic particle image velocimetry (SPIV) was employed in the present study for wake flow measurements in the cross-stream and streamwise directions. In the past, SPIV has been commonly used for wake measurements of yawed HAWTs (Bastankhah and Porté-Agel, 2016) and also for VAWTs (Rolin and Porté-Agel, 2018; Bensason et al., 2024). In the


**Figure 3.** (Left figure) Variation of angle of attack of the NACA airfoil for wind veer model of  $10^{\circ}$  (blue line) and  $20^{\circ}$  (red line). The dashed horizontal line represents the top and bottom extent of the disc, such that a total of  $10^{\circ}$  and  $20^{\circ}$  veer is generated from the two models, respectively; (Right figure) Wind veer model for generating a veer of  $10^{\circ}$  across the porous disc installed at the exit of the wind tunnel. A similar model that generates a wind veer of  $20^{\circ}$  (not shown here) is also tested.

present experiment, the seeding is done via a SAFEX smoke generator which releases smoke in the form of water-glycol fluid particles of average diameter of 1  $\mu$ m and a particle density of  $10^3$  kgm<sup>-3</sup>. The field of view (FOV) is illuminated by the Quantel *Evergreen* double-pulsed laser at a wavelength of 532 nm and delivering 200 mJ of energy per pulse. The thickness of the illuminated laser sheet is approximately 4 mm. Finally, images are captured using two LaVision sCMOS cameras (2560  $\times$  2160 px<sup>2</sup>, pixel pitch of 6.5  $\mu$ m/px) at a frequency of 15 Hz positioned on the opposite side of the laser sheet, as depicted in Fig. 2. A Scheimpflug adapter was used to adjust the focus of the camera plane to the measurement plane.

SPIV works on the principle of stereoscopic imaging, where two cameras simultaneously record the same image plane at different angles over two very closely spaced time intervals. The two views allow the extraction of out-of-plane motion of particles, along with the in-plane displacement of the tracer particles. Once the images are captured, they are divided into small interrogation windows, and cross-correlation techniques determine particle displacement within each window (Prasad, 2000). The resulting FOV is trapezoidal in shape, with a width and height of approximately 35.5 cm at a camera angle of  $78.84^{\circ}$ , using an AF Micro Nikkor lens of focal length 108 mm and numerical aperture of 8. A timestep of 45  $\mu$ s and 225  $\mu$ s between consecutive images is used for cross-stream and streamwise measurements, respectively. Table 2 summarizes the main parameters of SPIV employed in this study. A total of 100 vector fields were captured for each measurement plane and


Table 2. Important setup parameters of SPIV

| Parameters                        | SPIV setup                                    |  |  |
|-----------------------------------|-----------------------------------------------|--|--|
| Field of view (FOV)               | $353 \times 353 \text{ mm}^2$                 |  |  |
| Interrogation window size $(I_w)$ | $128 \times 128 \text{ px}^2$ for first pass, |  |  |
|                                   | $64 \times 64 \text{ px}^2$ for second pass   |  |  |
| Camera resolution (CR)            | $2560 \times 2160 \text{ px}^2$               |  |  |
| Image Resolution (IR)             | 7.04 px/mm                                    |  |  |
| Velocity fields (N)               | 100                                           |  |  |

averaged. Vector calculation was performed using a multi-pass stereo cross-correlation approach with decreasing interrogation window sizes. The first pass employed  $128 \times 128 \text{ px}^2$  ( $18 \times 18 \text{ mm}^2$ ) window size, followed by a second pass using  $64 \times 64 \text{ px}^2$  window size ( $9 \times 9 \text{ mm}^2$ ) with an overlap factor of 50 %.

The two cameras and laser are rigidly mounted on the traverse system with a spatial accuracy of 0.001 mm. The traverse system is capable of translational motion in both streamwise and spanwise directions. The spanwise measurements (i.e., wake cross-sections) at multiple downstream locations ranging from x/D=2 to x/D=8 are captured by moving the traverse system in the streamwise direction (shown in Fig. 1a). Meanwhile, streamwise wake measurements are taken across three different FOVs, with overlapping regions between consecutive imaging planes to ensure a smooth transition in wake propagation downstream, as illustrated in Fig. 1c. The streamwise FOVs are stitched together during the postprocessing of results by applying a Gaussian function in the overlapping region to ensure the gradients are smoothed out. It should be noted that the rotation stage that holds the porous disc is fixed at its location and is not mounted on the traverse system. It should be noted that a new calibration was performed after the experimental setup was rotated by  $90^{\circ}$  to measure streamwise velocity fields.

## 200 2.5 Flow measurement uncertainty

Following the work of Sciacchitano and Wieneke (2016), the uncertainty in velocity components, along with the derived quantities, is discussed in this section. Uncertainty quantification is crucial in PIV, particularly for an experiment as complex as this. It provides an estimation of the range that likely contains the true value of the variable of interest, thereby enhancing the reliability and interpretability of the results. The uncertainty in the time-averaged streamwise velocity can be expressed as:

$$U_{\bar{u}} = \frac{k\sigma_u}{\sqrt{N}} \tag{1}$$

where  $\sigma_u$  is the standard deviation of the streamwise velocity, k is the 95 % confidence interval i.e., 1.96, and N is the total number of instantaneous images (N=100). Similarly, uncertainties are calculated for in-plane velocity components (v and w). Moreover, the uncertainty in derived quantities, such as vorticity, can be expressed as follows:

$$U_{\omega_x} = \frac{U_{\bar{v} \text{ or } \bar{w}}}{d} \sqrt{1 - \rho(2d)} \tag{2}$$

**Table 3.** Uncertainty quantification of different variables on time-averaged data for three inflow cases. The percentages shown in the uncertainty quantification of velocity components are normalized with respect to the respective free-stream velocities averaged over the disc area. The values presented in this table represent the maximum observed values for each variable.

| Case                          | $U_u (ms^{-1})$ | $U_v (ms^{-1})$ | $\mathrm{U}_w~(\mathrm{ms}^{-1})$ | $\mathrm{U}_{R,uu}(m^2s^{-2})$ | $U_{TKE}(m^2s^{-2})$ | $U_{\omega_z} (s^{-1})$ |
|-------------------------------|-----------------|-----------------|-----------------------------------|--------------------------------|----------------------|-------------------------|
| Uniform inflow case           | 0.13 (1.26 %)   | 0.09 (0.87 %)   | 0.10 (0.97 %)                     | 0.30                           | 0.19                 | 14.68                   |
| Veered inflow of $10^{\circ}$ | 0.18 (1.57 %)   | 0.14 (1.22 %)   | 0.15 (1.31 %)                     | 0.53                           | 0.31                 | 22.84                   |
| Veered inflow of $20^{\circ}$ | 0.17 (1.56 %)   | 0.11 (1.01 %)   | 0.12 (1.10 %)                     | 0.38                           | 0.23                 | 17.94                   |

where,  $U_{\bar{v}\ or\ \bar{w}}$  is the uncertainty of mean in-plane velocity components, d is the grid spacing between the consecutive interrogation windows (here, d = 4.54 mm) and the cross-correlation factor  $\rho(2D)$  is approximated to be 0.45 (Sciacchitano and Wieneke, 2016). In addition to the uncertainties in velocity components and vorticity, uncertainty in Reynolds normal stress and turbulent kinetic energy (TKE) is also computed as follows:

$$U_{R,\mathrm{uu}} = \sigma_u^2 \sqrt{\frac{2}{N-1}} \tag{3}$$





$$U_{\text{TKE}} = \frac{1}{2} \sqrt{U_{R,\text{uu}}^2 + U_{R,\text{vv}}^2 + U_{R,\text{ww}}^2}$$
 (4)

where,  $\sigma_u$  is the standard deviation of streamwise velocity and TKE =  $\frac{1}{2}(\overline{u_i'u_i'}) = \frac{1}{2}(R_{uu} + R_{vv} + R_{ww})$ 

Table 3 summarizes the maximum uncertainty values for various variables across the three inflow conditions. As expected, the uncertainty in the veer inflow cases is slightly higher compared to the no-veer case, due to additional complexity in the flow introduced by vane-induced veer. Please note that the uncertainty values reported correspond to the shear region of the wake (at x/D = 5), where the flow is heavily separated.

#### 2.6 Flow characterization

For a complex system used to generate wind veer, such as the one employed in this study, it is essential to perform flow characterization in the absence of the porous disc. Ensuring a stable flow across all measurement planes is essential for obtaining reliable and consistent results in a wind tunnel experiment. Figure 4 shows the mean streamwise and spanwise velocities, along with turbulence intensity and wind veer variation for all the inflow conditions. The reference velocity ( $U_{\rm ref}$ ) used to normalize the velocity components is defined as the average velocity over the disc area located at x/D=0. For the uniform inflow case,  $U_{\rm ref}=10.3~{\rm ms}^{-1}$ . It is important to note that the placement of veering vanes at the tunnel exit induces a slight flow acceleration downstream of the vanes. As a result, for the veered inflow of  $10^{\circ}$ ,  $U_{\rm ref}=11.43~{\rm ms}^{-1}$ , while for the veered inflow of  $20^{\circ}$ ,  $U_{\rm ref}=11.04~{\rm ms}^{-1}$ .

As expected, the streamwise velocity profile for the clean (uniform inflow) case remains consistent across all downstream locations (see Fig. 4a). A minor spanwise velocity component of approximately 2 % is observed, likely caused by a slight

**Figure 4.** Flow characterization at different downstream locations in the absence of porous disc at y = 0: (a), (b), (c) - each of these figures show streamwise, spanwise, and turbulence intensity variation for the clean case, wind veer model of  $10^{\circ}$  case, wind veer model of  $20^{\circ}$  case, respectively; (d) - Wind veer variation for all the cases. The dashed horizontal line at z/D = -0.5 and 0.5 indicates the span area of the porous disc.

misalignment of the laser sheet with the wind tunnel exit. The turbulence intensity within the disc region remains below 1 % at all streamwise planes, indicating highly stable flow conditions in the measurement domain.



For the case of  $10^{\circ}$  wind veer model, the actual total veer at the porous disc location (x/D=0) is  $8.91^{\circ}$ . This value drops down to  $6.97^{\circ}$  at x/D=7. The veering vanes induce spanwise velocity in the flow that varies across the porous disc: positive above the centerline and negative below. However, the presence of the vanes results in a non-uniform streamwise velocity distribution across the disc. The minimum streamwise velocity typically deviates by less than 5 % from the mean reference velocity across all downstream locations. As the influence of the veering vanes diminishes downstream, the streamwise velocity gradually recovers and approaches the free-stream value. As anticipated, the veer configurations increase turbulence intensity due to partial flow blockage induced by the vanes. Despite this, the turbulence intensity across the disc area remains below 2 % at all measured streamwise locations for both veered cases, indicating relatively low levels of added turbulence by the veering vanes. The effect of turbulence on wake recovery is isolated in the momentum budget analysis discussed in Section 3.3.

For the 20 ° wind veer configuration, the actual veer angle is  $14.3^{\circ}$  at x/D=0, reducing to  $12.08^{\circ}$  at x/D=5. In the wind veer plots (Figure 4d), the magenta dashed line denotes the ideal (target) veer profile. The root mean square error (RMSE) between the target and measured veer profiles lies within the range of  $1.23^{\circ}$  to  $1.53^{\circ}$  for both veer cases, within the vertical range of  $z/D \in [-0.5, 0.5]$ . Although the reduction in wind veer downstream does not perfectly replicate real conditions, this deviation does not substantially affect the qualitative behavior of the wake evolution. An additional discussion of flow characterization by visualizing contours in the horizontal plane (x-y) is given in Appendix A.

# 250 3 Results and discussion

## 3.1 Streamwise velocity fields

The normalized streamwise velocity contours corresponding to three different inflows—no-veer, veer  $10^{\circ}$ , and veer  $20^{\circ}$ —under two different yaw angles of  $0^{\circ}$  and  $30^{\circ}$  are presented in Fig. 5, Fig. 6, and Fig. 7, respectively. Each figure presents the inflow velocity field to the disc and a three-dimensional representation of wake evolution, where cross-stream planes at x/D=3, 5, and 7 are overlaid on the streamwise plane, for yaw angles of  $0^{\circ}$  and  $30^{\circ}$ . The figures also include planar views of streamwise velocity contours with in-plane velocity vectors at x/D=5 for both yaw angles. As the wakes are not axisymmetric for veered inflow, it is of interest to find the location of the wake center and how it moves downstream for both non-yawed and yawed cases. Here, the wake center is computed at each cross-stream plane (x/D=3, 5, and 7) using the center of mass method (Howland et al., 2016) using the following equations:

$$y_c(x) = \frac{\int \int y \Delta u(x, y, z) dy dz}{\int \int \Delta u(x, y, z) dy dz}; z_c(x) = \frac{\int \int z \Delta u(x, y, z) dy dz}{\int \int \Delta u(x, y, z) dy dz}$$
 (5)

In equation 5,  $y_c(x)$  and  $z_c(x)$  are the wake centers in the spanwise and vertical directions, and the streamwise velocity deficit is defined as  $\Delta u(x,y,z) = U_{\infty} - U_x(x,y,z)$ . The integration is performed at each cross-sectional plane. The wake center for different inflows and yaw angles of  $0^{\circ}$  and  $30^{\circ}$  is shown in Fig. 8.

For the no-veer and non-yawed disc case, the wake exhibits a nearly circular and symmetric profile in the region not influenced by the tower. As the tower is relatively thicker than the diameter of the porous disc (with a disc-to-tower diameter ratio of 10), it induces a small vertical transport of momentum pointing downwards (also evident from the in-plane vectors in Fig.






5b behind the tower, where arrows point downwards), which makes the wake asymmetric and shifts the wake center downward toward the lower half of the disc (also see Fig. 8a). This was also observed in wind tunnel studies by (Pierella and Sætran, 2017; Schottler et al., 2018). As the wake advects downstream, as expected, it expands in both lateral and vertical directions due to flow entrainment from the free-stream. Notably, across all cross-stream planes in the non-yawed case under uniform inflow (see Fig. 5b), the wake consistently exhibits a lateral displacement toward the left. This is likely because the laser sheet is not perfectly parallel to the wind tunnel exit plane and is slightly misaligned. Supporting evidence for this is found in flow characterization measurements conducted without the porous disc (see Fig. 4 for reference), which also reveal a small spanwise velocity component of approximately 2 % of the free-stream velocity in the disc region. Furthermore, the relatively thick tower may contribute to this shift by acting as a bluff body that sheds vortices, thereby introducing asymmetries through complex interactions between the tower and disc wakes. As this lateral shift is systematic and consistent across all cases, it does not qualitatively or quantitatively impact the comparative analysis and conclusions presented in this study.

In the yawed case for no-veer inflow (Fig. 5c), the characteristic curled (kidney-bean-shaped) wake was observed. This shape arises from the lateral force exerted by the disc on the incoming flow, which induces a significant spanwise velocity in the wake. As a result, two counter-rotating vortices form, originating from the top and bottom edges of the disc. These findings are consistent with previous experimental works (Howland et al., 2016; Bastankhah and Porté-Agel, 2016; Hulsman et al., 2022), where turbines or discs were immersed in uniform boundary-layer inflow. As evident from the streamwise plane velocity contours, the wake deflects sideways and recovers faster when the turbine is yawed. This is primarily due to the reduced thrust coefficient of the porous disc under yaw. The wake width in the horizontal xy-plane also appears thinner for the yawed disc because the mean spanwise velocity induced by yaw deflects and pushes the wake core sideways—a result that is also consistent with prior studies (see, for instance, Bastankhah and Porté-Agel (2016) and Schulz et al. (2017)). Also, it is quite obvious that yawing the disc moves the wake center in the opposite direction of yaw as seen in Fig. 8a, with maximum lateral movement of the wake center at the farthest downstream location. Therefore, yaw induces asymmetry in wake in both spanwise and vertical directions. As expected, the mean streamwise velocity is observed to be higher (i.e., the wake deficit is smaller) for the yawed case.

Figure 5. Normalized streamwise velocity wake contours for the uniform inflow case for yaw angles of  $0^{\circ}$  and  $30^{\circ}$ . For reference, the inflow at the disc location (x/D=0) is shown in the top panel. The middle and the bottom panels present combined cross-sectional planes overlaid on the streamwise plane for the yaw angles of  $0^{\circ}$  and  $30^{\circ}$ , respectively. The streamwise velocity contour at the cross-stream location x/D=5 is also included, with quivers indicating the in-plane velocity vectors. The solid circle represents the disc span, while the yawed disc is shown by a dashed line. The dotted black line depicts the wake boundary, defined as the contour where  $U_x/U_{\infty}=0.9$ .

Figure 6. Normalized streamwise velocity wake contours for the veered inflow case with a  $10^{\circ}$  veer. The rest of the caption is the same as Fig. 5.

The wake topology in veered inflow cases is distinctly different from the uniform inflow, as seen by comparing the contours in Fig. 5 and Fig. 6. As a result of the veer in the inflow, the wake shape appears skewed in the lateral direction that extends






moving along in the streamwise direction. This can be attributed to the variation of spanwise velocity with height—positive above the disc and negative below—as also illustrated by the spanwise velocity components in Fig. 4b. Consequently, the wake stretches in the +y-direction above the disc and -y-direction below it. The elliptical wake shape for the veered inflow observed agrees well with the existing numerical simulation studies on stably stratified flows (Abkar and Porté-Agel, 2016; Vollmer et al., 2016; Churchfield and Sirnivas, 2018; Klemmer and Howland, 2024; Narasimhan et al., 2022, 2025). When the disc is yawed, the curled shape due to yaw is superimposed on the elliptical wake shape due to veer, resulting in a complex wake structure that helps direct the flow away from the disc area, thereby exposing the downwind turbine to the higher free-stream velocity. Under veered inflow conditions, the tower wake in Fig. 6b shows that the velocity deficit caused by tower blockage can be displaced by up to 1D to the right, due to the lower-half veer pointing in that direction. At farther downstream locations, it merges with the wind turbine wake due to turbulent mixing. As pointed out in earlier studies (Santoni et al., 2017; Abraham et al., 2019), tower wake is important as it influences wake dynamics and interacts with surface fluxes.

As the magnitude of wind veer in the inflow increases (see Fig. 7), the wake skews even further due to higher spanwise velocity, characterized by an extended major axis and a reduced minor axis. This means that the velocity deficit is concentrated more in a narrow bend along the minor axis direction. As hypothesized in Churchfield and Sirnivas (2018), this effect can accelerate wake recovery as the free-stream flow now has to travel a shorter distance to reach the wake core. A higher veer in the inflow results in wake thinning, which leaves more undisturbed free-stream wind speed for downwind turbines, potentially leading to more power available for them. An interesting observation from these results is that as the magnitude of wind veer increases, it exerts the dominant control on the wake shape, reducing the relative influence of yaw.

To better quantify wake recovery under veered inflow, Fig. 9 presents heat maps of the maximum wake deficit in the wake region across various cross-sectional planes for different yaw angles and veer inflows. The no-veer baseline behaves as expected: as yaw increases, the peak deficit falls monotonically, indicating accelerated wake recovery under yawed conditions. It is evident from the wake deficit heatmap that veer inflow results in lower maxima than the baseline case at every downstream location, confirming that veer drives the dominant recovery effect. Interestingly, within each veer case, increasing yaw from  $0^{\circ}$  to  $30^{\circ}$  does not alter the peak deficit significantly at that plane. This indicates that the effect of yaw on wake recovery diminishes compared to veer. Instead, yaw primarily displaces the wake laterally, leaving the recovery rate governed mostly by the veer strength.

A quantitative comparison reinforces this observation: under a  $10^{\circ}$  wind veer, the non-yawed disc exhibits reductions in wake deficit of 15%, 32%, and 39% at x/D=3, 5, and 7, respectively, relative to the baseline case. In contrast, for the yawed disc under the same veer, the reductions are substantially smaller—7%, 5%, and 17%, respectively. This highlights a key finding: if wind veer is high in the atmosphere under stable conditions, as is typically the case, as reported in multiple field experiments, yawing the upstream wind turbine may not be advantageous for wake steering purposes. Since wake recovery is already enhanced by wind veer, there appears to be limited value in applying yaw control solely for the purpose of maximizing downstream wind turbine power production. This will be further discussed in Section 3.5.

Figure 7. Normalized streamwise velocity wake contours for the veered inflow case with a  $20^{\circ}$  veer. The rest of the caption is the same as Fig. 5.

**Figure 8.** Wake center for three inflow cases corresponding to yaw angles of  $0^{\circ}$  (represented as circles) and  $30^{\circ}$  (represented as crosses). The red, green, and blue colors represent the cross-stream plane location (x/D) of 3, 5, and 7, respectively.

Figure 9. Wake deficit heat map for disc under yaw angles of  $0^{\circ}$ ,  $10^{\circ}$ ,  $20^{\circ}$ , and  $30^{\circ}$ , at different downstream cross-sectional planes of x/D=3, 5, and 7 under  $0^{\circ}$ ,  $10^{\circ}$ , and  $20^{\circ}$  veered inflow.

#### 3.2 Vorticity fields


The measured time-averaged streamwise vorticity contours for the three inflow cases at cross-stream location x/D=5 for the yaw angle of  $30^{\circ}$  are shown in Fig. 10a. As also discussed in Section 3.1, in the baseline case without veer, the curled wake shape observed behind the yawed disc arises from the formation of counter-rotating vortex pairs (CVP), which shift the wake laterally. A slight asymmetry in the vorticity distribution of top and bottom vortices for this case can be attributed to the relatively thicker tower that sheds its own vortex, which is not negligible. This tower-induced vortex merges with the bottom vortex of the disc as the wake propagates downstream.

In contrast, veered inflow leads to increased stretching and distortion of streamwise vorticity contours, with the effect intensifying under stronger wind veer. It can also be noticed from Fig. 10a that the vorticity strength in veered inflow is significantly






affected by background veer vorticity: the strength of the top vortex decreases compared to the corresponding top vortex in the uniform inflow case with no-veer. These observations are consistent with the numerical results of (Narasimhan et al., 2022) and further supported by the variations in maximum streamwise vorticity and the circulation of the top vortex as shown in Fig. 10b and 10c. Circulation in the top vortex was computed by spatially integrating the positive vorticity field over that region  $(\Gamma = \int \int \omega_x dy dz)$  following the approach highlighted in Xu et al. (2025). In general, with increasing downstream distance, the peak vorticity strength for the yawed cases under veered inflow is lower than in the baseline yaw case. This can be explained as follows: because the inflow already contains nonzero streamwise vorticity with opposite orientation to the upper vortex, the background veer reduces its strength. Consequently, higher veer cases accelerate the decay of circulation in the CVP (Fig. 10c, with lower values than the yaw case under uniform inflow. As expected, due to symmetric pressure distribution, circulation remains relatively constant downstream for the non-yawed disc under uniform inflow. In contrast, lateral shear induced by veer in the inflow results in a non-symmetric pressure distribution around the disc, leading to a decrease in circulation with downstream distance. Interestingly, when the disc is vawed in the baseline case, circulation slightly increases from x/D=2to x/D=3, before continuing its downward descent, as also seen in (Shapiro et al., 2020). In contrast, the circulation for the veer inflow of  $10^{\circ}$  and yawed disc is nearly constant from x/D=2 to 3, before decaying further downstream. Whereas, the circulation in the veer inflow of 20° decreases monotonically. This again highlights the dominant role of background veer in reducing circulation strength, which increasingly overshadows yaw effects as veer magnitude grows.

Additionally, the vertical vorticity contours  $(\omega_z = \frac{\partial v}{\partial x} - \frac{\partial u}{\partial y})$  shown in Fig. 11 indicate that veered inflow accelerates vortex sheet breakdown compared to the no-veer case. This is evident from the golden contour lines marking regions where  $\omega_z D/U_\infty = \pm 1$ . For the no-veer case, the peak vorticity persists beyond a downstream distance of 6D, whereas in the veered inflow it disappears within 5D. This faster decay is attributed to stronger lateral (cross-flow) velocity components in the wake introduced by veer. The decay of vorticity is slightly faster when yaw is superimposed on veer, which results in additional spanwise velocity in the wake. For the yawed disc immersed in the wind veer of  $20^\circ$ , the counterclockwise (CCW) vortex bifurcates into two, visually consistent with the wake topology discussed in Section 3.1. Overall, these results suggest that increasing veer intensity leads to faster CVP decay and enhances turbulent mixing in the wake, thereby promoting faster wake recovery.

Figure 10. (a) Streamwise vorticity contours for three different inflows for  $30^{\circ}$  yawed disc at x/D = 5; (b) Streamwise distribution of max vorticity and circulation measured for the upper vortex (rotating CCW) for different veer inflows and yaw angles of  $0^{\circ}$  (represented as dashed lines) and  $30^{\circ}$  (represented as solid lines)

# 360 3.3 Momentum budget analysis

In this section, dominant mechanisms driving wind turbine wake recovery as a result of yaw and veered inflow conditions are analyzed via the momentum budget analysis of the Reynolds-averaged Navier-Stokes (RANS) equation in the streamwise direction. This analysis helps to understand how momentum is redistributed in the wake. The time-averaged RANS equation

Figure 11. Time-averaged vertical vorticity  $(\omega_z)$  contours on the horizontal plane. Figures on the left are for the non-yawed disc, whereas figures on the right are for the yawed disc of 30°. The golden lines on the contour represent regions where  $\omega_z D/U_\infty = \pm 1$ .

in the streamwise direction can be written as:



365 
$$\bar{u}\frac{\partial \bar{u}}{\partial x} = -\underbrace{\bar{v}\frac{\partial \bar{u}}{\partial y}}_{I} - \underbrace{\bar{w}\frac{\partial \bar{u}}{\partial z}}_{II} - \frac{1}{\rho}\frac{\partial p}{\partial x} - \frac{\partial(\overline{u'u'})}{\partial x} - \underbrace{\frac{\partial(\overline{u'v'})}{\partial y}}_{III} - \underbrace{\frac{\partial(\overline{u'w'})}{\partial z}}_{IV} - \underbrace{\frac{\partial(\overline{u'w'})}{\partial z}}_{IV}$$
 (6)

In the above equation, u, v, and w denote the streamwise, spanwise, and vertical velocity components, respectively. The overbar indicates the time-averaged values of these components, while the primes represent instantaneous velocity fluctuations. The term I in Eq. 6 denotes cross-stream advection that represents the transport of streamwise momentum by the mean spanwise velocity  $(\bar{v})$ . The term II represents the transport of streamwise momentum in the vertical direction by the mean vertical velocity  $(\bar{w})$ . Lastly, terms III and IV are the divergence of the cross-stream Reynolds stress  $(\bar{u'v'})$  and the vertical Reynolds stress  $(\bar{u'v'})$ , which represent turbulent mixing of momentum laterally and vertically, respectively. It should be noted that the variation of streamwise Reynolds stress gradient on the RHS is not computed because of sufficiently large spacing between spanwise planes. Similarly, the pressure gradient term is also neglected as it is not measured in the experiments. Viscous terms are also neglected due to the high Reynolds number of the flow. Figure 12 represents the contributions of the remaining terms in Equation 6 at X/D=5 for six different cases. The wake edge is represented by a dotted line. The positive (red) region indicates favorable contributions to wake recovery, and vice versa.

The contours in the first vertical column in Fig. 12 show the mean lateral advection of streamwise momentum. It can be seen that for the no-veer inflow, the positive values are concentrated along the right edge of the wake, and vice versa. For the yawed disc in the no-veer case, due to significant spanwise velocity induction into the wake, the positive values are much higher than in the non-yawed cases. Interestingly, it can be seen visually that the positive values for the yawed disc are slightly lower for the veered cases compared to the uniform inflow case. This could be attributed to the spanwise component of background veer counteracting the spanwise flow induced by yaw. In the no-veer case, the low-momentum wake is ejected out laterally from the center to the left side of the wake edges (blue region). In contrast, for the veered cases, this ejection takes place from both the left and right edges of the wake due to the varying spanwise velocity with height, whose sign also changes. Moreover, comparing the relative contributions of all terms in the second row of Fig. 12, it is obvious that the advective terms play a key role in redistributing momentum for the yawed case compared to the divergence of shear stress terms.

The contours in the second column show the mean vertical advection of streamwise momentum. The contribution of term II to wake recovery is small for the uniform inflow case relative to the veered inflow case. In the latter, a large red region is present along the upper edges of the wake. This can be attributed to the fact that, due to the skewed wake shape, the gradients in the shear layer are sharper, making the injection of free-stream flow into the wake easier.

The contours in the third column show the Reynolds stress term distribution in the lateral direction. A visual comparison of contours of term III for all cases reveals that its contribution remains similar across all cases.

Lastly, the contours in the fourth column show the Reynolds stress term distribution in the vertical direction. It can be seen visually that, compared to the baseline case, the contribution of the streamwise-vertical Reynolds stress term  $(-\frac{\partial u'w'}{\partial z})$  to the overall budget is higher in the veered case compared to the uniform inflow case, where both its magnitude and area are relatively small. The positive and negative values are concentrated in the wake core and wake edges, respectively, which means more momentum is entrained from the top and bottom in the veered inflow scenario. This can be attributed to the fact that velocity gradients are sharper due to the elliptical shape of the wake.

To evaluate the overall impact of the momentum budget terms on wake recovery across different cross-stream planes, we present their individual contributions to the recovery process at each plane. The Eq. 6 can be rewritten as:

$$\frac{\partial \bar{u}}{\partial x} = \frac{1}{\bar{u}} \left( -\bar{v} \frac{\partial \bar{u}}{\partial y} - \bar{w} \frac{\partial \bar{u}}{\partial z} - \frac{1}{\rho} \frac{\partial \bar{\rho}}{\partial x} - \frac{\partial \bar{u}' u'}{\partial x} - \frac{\partial \bar{u}' v'}{\partial y} - \frac{\partial \bar{u}' w'}{\partial z} \right) \tag{7}$$

where  $\frac{\partial \bar{u}}{\partial x}$  is also called the streamwise wake recovery rate. Fig. 13 shows the bar plot of each term net contribution to wake recovery, which is integrated within the wake region at three different downstream locations of x/D=3, 5, and, 7 for three different inflows and two yaw angles of  $0^{\circ}$  and  $30^{\circ}$ . At x/D=3 for the no-veer case, the lateral advection of streamwise momentum (Term I) helps in wake recovery more compared to the other terms. This term is slightly higher for the yawed cases because of the induction of spanwise flow in the wake, which contributes positively to the wake recovery. At further downstream locations, the contribution of this term reduces in the overall wake recovery. For the veered inflow and no-yaw case, the vertical advection of streamwise momentum (Term II) plays a dominant role in redistributing momentum in the wake, as evident from the sharp peaks of term II at x/D=3. The term II in wind veer of  $20^{\circ}$  is approximately 5 times higher than the corresponding term in the baseline case. However, when the turbine is yawed, term II decreases with a corresponding

Figure 12. Measured terms of Eq 6 at X/D=5. The disc projection is represented by a solid circle, whereas the projection of the yawed disc is shown as a dashed line. The wake contour is shown as a dotted line where  $U_x/U_\infty=0.9$ . The asterisk indicates that each term is normalized by the factor  $D/U_\infty^2$ .

Figure 13. Terms of streamwise momentum budget equation 7 integrated over the wake region defined as where  $Ux/U_{\infty}=0.9$ . Each term is shown at three different cross-stream locations x/D=3, 5, and, 7 for three different inflows and yaw angles of  $0^{\circ}$  and  $30^{\circ}$ .

increase in term I. This means that more entrainment of fluid occurs into the wake from the vertical direction than the spanwise direction.

## 3.4 Turbulent kinetic energy

Turbulence kinetic energy (TKE) reflects the energy content of turbulence in the wake and how it is spatially distributed. It is an important parameter influencing wind turbine performance and blade fatigue loads. The wake-added TKE is computed as follows:

$$\Delta TKE = TKE_{with \ disc} - TKE_{without \ disc} \tag{8}$$

where  $TKE = \frac{1}{2}\overline{(u_i'u_i')}$ . In this equation,  $u_i'$  is the fluctuating velocity component in the *i*-th direction (where i = 1, 2, 3 corresponds to u, v, w) and  $\overline{u_i'u_i'}$  is the time-averaged Reynolds stress tensor components.

**Figure 14.** Distribution of wake-added turbulent kinetic energy for the three different inflows. The figures on the left correspond to the no-veer inflow, the middle figures to the  $10^{\circ}$  veer inflow, and the right figures to the  $20^{\circ}$  veer inflow. The figures in the top panel are for the yaw angle of  $0^{\circ}$ , whereas the figures in the bottom panel are for the yaw angle of  $30^{\circ}$ . The solid circle on the cross-stream plane represents the projection of the non-yawed disc.

Figure 14 shows the streamwise and cross-section planes of wake-added TKE for non-yawed and yawed discs for the three inflows. In the near wake, the TKE is enhanced in the lower half of the disc region due to the strong influence of vortices shed by the tower and its interaction with the wake shed by the porous disc. TKE distribution is ring-shaped in the near wake, becoming more uniform at farther downstream distances due to flow entrainment and wake mixing. The wake-added TKE is highest for the veered inflow cases compared to the baseline no-veer case. Yaw introduces asymmetry in the TKE field, with more TKE addition from one side of the wake, as evident from the streamwise plane of TKE in Fig. 14. Similar to the velocity wake contours, the wind veer has an effect of stretching the TKE laterally. A larger TKE in the wake also implies a higher degree of wake mixing, thus aiding in wake recovery. The TKE distribution in the horizontal plane (x – y) under veered inflow and non-yawed conditions is markedly different from that in the baseline case, with the TKE expanding more in the lateral direction for the veered inflow.

The increase in TKE for the veered cases can be explained by looking at the profiles of shear production of turbulence  $(\mathscr{P} = -\overline{u_i'u_j'}\frac{\partial \overline{u_i}}{\partial x_j})$  in the lateral cross-sectional plane, which is the dominant term in the TKE budget equation, as shown in Fig. 15. In this equation,  $\overline{u_i'u_j'}$  are the Reynolds stresses,  $\frac{\partial \overline{u_i}}{\partial x_j}$  is the mean velocity gradient. It is evident that the shear production of

Figure 15. Distribution of shear production of turbulence. The dotted black line depicts the wake shape defined as where  $U_x/U_\infty=0.9$ .

turbulence is much higher in the veered inflow compared to the no-veer case. This is due to higher lateral shear in the former case than in the latter. Wind shear (both vertical and lateral) is one of the driving mechanisms for turbulence generation along with atmospheric stratification.

In our study, as we only have lateral wind shear, this is the main source of turbulence production. In the near wake (x/D = 3), the high production of TKE is mainly concentrated near the left and right wake edges. As velocity gradients decrease





downstream due to mixing and wake recovery (at x/D=5), the production term drops. However, at the same location, the wake-added TKE is still elevated due to accumulated turbulence (see Fig. 14). It should be noted that in the wake core region at x/D=3, the TKE production is almost negligible, which is mainly due to very low velocity shear in the wake core.

# 3.5 Available power

The analysis until now focuses on the mean wake deficit and turbulence quantities of porous disc wake under uniform and veered inflows, and the dominant terms in the wake recovery mechanism. To further highlight the role of wind veer inflow in replenishing kinetic energy in the wake, an analysis of available power (AP) is performed at selected downstream locations. Fig. 16 shows the available power for different cases at two downstream locations x/D = 5 and 7. The AP is calculated using the relation given in Vollmer et al. (2016) and Zong and Porté-Agel (2020):

$$f_{AP}(x_T, y_T, z_T) = \frac{\iint_G U(x_T, y, z)^3 \, dy \, dz}{\iint_G U_{\text{in}}(y, z)^3 \, dy \, dz} \tag{9}$$

where  $f_{AP}(x_T, y_T, z_T)$  is the normalized available power at the turbine location,  $U(x_T, y, z)$  is the wind speed at a given point in the wake, and  $U_{in}(z)$  is the wind speed at the location of the porous disc. The integration is performed over the region G, defined as

$$(y - y_T)^2 + (z - z_T)^2 \le R^2, (10)$$

which represents a circular area of radius R centered at hub height  $z_T$  and lateral position  $y_T$ .

For the no-veer inflow and non-yawed disc condition (solid blue line in Fig. 16,  $f_{AP}$  is lowest among all other cases, increasing from 32 % at x/D=5 to 47 % at x/D=7 for an inline configuration with the downstream turbine positioned at y/D=0. When the disc is yawed (dashed blue line), the wake center shifts to the left, and the overall available power increases. The lowest value of  $f_{AP}$  for the yawed case is 33 % higher than the non-yawed case. In the presence of veer in the inflow and a non-yawed disc (solid red and green lines), a stronger veer (20°) results in more available power than a weaker veer (10°). This is expected, as veer results in a stretched and elongated wake structure, exposing the downstream turbine to higher free-stream velocity inflow and thus leading to higher AP. To quantify this, if a hypothetical turbine is placed at x/D=5, the AP for it under veered inflow of 20° is 45 % more compared to when the turbine is operating in no-veer inflow. When yaw is introduced in the presence of veered inflow (dashed red and green lines), wake recovery is enhanced even further because yaw deflects the wake farther away in the positive y-direction, and  $f_{AP}$  increases to 77 %. Similar trends are also observed at the downstream location of x/D=7. At this location, the highest gains in available power up to 85 % can be observed for the veered inflow of  $20^{\circ}$  and a positive yaw steering of  $30^{\circ}$ .

To investigate optimal downwind turbine placements with lateral offsets, Fig. 17 presents contour maps of the available power coefficient for all cases, computed using a sliding integration window across the spanwise plane. The dashed and solid lines on the contours denote locations where the coefficient of available power,  $f_{AP}$ , is 0.50 and 0.75, respectively. A similar analysis for vertical-axis wind turbines has been conducted by Bensason et al. (2025).

Figure 16. Coefficient of available power at two downstream locations (a) x/D = 5 and (b) x/D = 7 for different cases. The legend notation is  $WV\phi Y\theta$ , where  $\phi$  denotes the wind veer angle (in degrees) and  $\theta$  denotes the yaw angle (in degrees). For example, WV10Y30 corresponds to a  $10^{\circ}$  wind veer inflow and a  $30^{\circ}$  yaw misalignment of the disc.

In the baseline case with no-veer, the available power deficit exhibits a lateral shift to the left with increasing yaw angle. As discussed previously, yawed turbines generate reduced  $C_T$ , resulting in greater  $f_{AP}$  values further downstream. However, a downwind turbine positioned directly inline—even at x/D=7 from the upstream—would still experience a partial wake, potentially exacerbating structural loading. In such scenarios, a more favorable placement would involve a slight negative lateral offset to avoid the lower wake regions of the upstream turbine.

In contrast, veer in the inflow leads to a more laterally uniform distribution of the available power deficit, particularly at x/D = 5 and x/D = 7. This is evident from the absence of the  $f_{AP} = 0.5$  contours (dashed lines) at x/D = 7, indicating that  $f_{AP}$  exceeds 0.5 across the entire span. A stronger veer of  $20^{\circ}$  further enhances this uniformity, with the dashed lines disappearing even at x/D = 5. Under this condition, the  $f_{AP}$  contours for yaw angles of  $0^{\circ}$  and  $30^{\circ}$  appear nearly identical.

This observation is particularly noteworthy, as wake steering strategies typically perform best under low-turbulence conditions, which are characteristic of stable atmospheric boundary layers. Wind veering is more prevalent under such stratified conditions. Therefore, even without active yaw control, the wake may have sufficiently recovered by the time it reaches a downwind turbine located at x/D = 7. This recovery could be beneficial for dense wind farm configurations, where turbine spacing often falls within the range of x/D = 5 to 7.

#### 4 Conclusions



The study presents the first experimental investigation into the effect of wind veer on wakes behind a porous disc. Three different inflow conditions were examined: no-veer, a veer of 10°, and a veer of 20°, each tested across a range of disc yaw



**Figure 17.** Filled contour of available power distribution for all cases at three different downstream locations. The solid vertical line represents the locations where  $f_{AP}$  is 0.75 and the dashed vertical line represents locations where  $f_{AP}$  is 0.5.

angles. Stereoscopic PIV was adopted to study the velocity fields in the wake. The flow characterization without the porous disc demonstrates that our experimental setup can effectively generate wind veer and sustain it reasonably well, even at far downstream distances up to x/D=7 in the wind tunnel. Contours of streamwise velocity and vorticity reveal that, in the absence of wind veer, the wake of a yawed disc exhibits a distinct curled shape. This structure is attributed to the formation of counter-rotating vortex pairs (CVPs) shed from the edges of the disc. Wind veer results in a skewed wake shape that resembles an ellipse, which stretches even further for a higher wind veer of  $20^{\circ}$ . The combined action of veer and yaw results in a complex-shaped wake, with veer effects dominating yaw in determining wake shape as veer in the inflow increases. Background wind veer significantly distorts streamwise vorticity and makes the vortices asymmetric. Both the peak vorticity and circulation for veered inflow are lower than in the no-veer inflow case. The vertical vorticity contour reveals that the vortex sheet dissipates faster under veered inflow due to significant cross-flow and enhanced turbulent mixing. This is also observed in the turbulent kinetic energy (TKE) contours, which show higher wake-added TKE for the disc immersed in veered inflows. This is related to the higher shear production of turbulence in the wake as a result of lateral wind shear under veered inflow, which further enhances wake recovery. The analysis of the terms of the RANS budget equation reveals the differences in the dominant wake recovery mechanisms for the uniform inflow and veered inflow. The mean lateral advection of momentum is dominant in uniform inflow for wake recovery, whereas the mean vertical advection of momentum is the main driving mechanism in



redistributing momentum for turbines in veered inflow. Furthermore, for the veered inflow, the contribution of the divergence of the vertical shear stress term is higher than that of the lateral shear stress term, revealing that sharper velocity gradients, as a result of the skewed wake shape, entrain more free-stream momentum into the wake.

The analysis of available power in the wake for different inflows and yaw angles provides interesting insights regarding the turbine control strategies, such as wake steering. It was observed that veered inflow exhibits more uniform available power coefficients in the wake compared to the no-veer case, even when the disc is not yawed. This means that for a hypothetical turbine operating in the wake of the upwind turbine, it experiences higher available power throughout the spanwise distance, reducing the probability of the downwind turbine operating in the partial wake scenario, thereby reducing the fatigue and fluctuating loads on the turbine. In fact, for a rotor positioned inline of the upwind turbine at x/D = 5, the available power coefficient is 45 % greater for a 20° veered inflow than for the baseline case with uniform inflow under non-yawed conditions. Under the combined action of wind veer and yaw, as expected, wake recovery is accelerated even further. Although the available power increases compared to the no-yaw case for an inline downwind turbine when the upstream rotor is yawed, the relative improvement is modest. For instance,  $f_{AP}$  for a second turbine is only 9 % more for the veered inflow of 20° compared to the no-veer inflow for a yaw angle of 30°. These results highlight that under moderate wind veer conditions, such as those tested in this experiment, yawing a turbine may not result in significantly larger benefits for power production.

The presented results highlight the importance of considering the effect of wind veer for designing wake-steering-based control strategies. Ignoring wind veer in look-up tables for turbine yaw angles for wake steering could result in sub-optimal performance of overall wind farms and also increase structural loading on downstream wind turbines. Future experimental work should involve considering the influence of turbine rotation and ground effect on wake characteristics.

## 520 Appendix A: Flow characterization in horizontal planes

The mean spanwise and vertical velocity contours in the absence of the porous disc on the horizontal plane are shown in Fig. A1. For the no-veer inflow, there is a slight positive spanwise velocity in the horizontal plane. Due to veer in the inflow, the direction of the spanwise velocity points in the negative y-direction, with magnitude increasing as wind veer increases. On the other hand, the vertical velocity component for the no-veer inflow is almost negligible. However, due to veer, there is a vertical velocity component in the wind, with its direction positive on one side (i.e., coming out of the plane) and negative on the other side (i.e., going into the plane). This is mostly due to enhanced turbulence in the veered inflow and the slight induction of vertical motions resulting from the veer generation method employed in this study.

# Appendix B: Spanwise and vertical velocity fields

To complement the discussion in Section 3.1, it is useful to examine how wind veer inflow influences mean spanwise and vertical velocity in the wake. Figure B1 presents the spanwise and vertical velocity fields in the horizontal plane for the non-yawed (top panel) and yawed porous disc of 30° (bottom panel). For the non-yawed case, the spanwise velocity field resembles



**Figure A1.** Flow characterization contours in the absence of the porous disc, showing the spanwise and vertical velocity components for the three inflows

the distribution observed in the flow characterization results shown in Fig. A1. This indicates that, under veered but non-yawed conditions, the presence of the porous disc does not induce any significant spanwise flow into the wake, and the spanwise velocity is predominantly determined by the incoming veered inflow.

In contrast, the vertical velocity component shows pronounced differences. For veered inflow under non-yawed conditions, wake rotation is visible, whereas, as expected, no rotation is observed for uniform inflow without yaw. Following Bastankhah and Porté-Agel (2015), wake rotation can be identified when vertical velocity components of opposite sign appear in the horizontal plane. Under yawed conditions, the vertical velocity contours reveal further notable features: compared to clean inflow, veered inflow induces a net downward transport of momentum from above, enhancing wake recovery. This behavior is consistent with the momentum budget analysis in Section 3.3.

The spanwise velocity distribution for the yawed disc also shows pronounced differences. As seen in the discussion of Section 3.1, a yawed disc exerts a lateral force on the flow, which in return induces spanwise velocity in the wake (for our case, in the positive *y*-direction), which is clearly visible for the no-veer case. Interestingly, as the veer acts in the opposite direction to the spanwise velocity, it counteracts and diffuses the spanwise velocity induced by turbine yaw. As the degree of veer across the disc increases, the yaw-induced spanwise velocity decreases further, as evident in Fig. B1. At higher veer magnitudes, the influence of veer clearly dominates over the yaw effect. This underscores the importance of accounting for wind veer in wake steering control strategies.

Figure B1. Spanwise and vertical velocity components in horizontal plane for no-yaw and yaw conditions

https://doi.org/10.5194/wes-2025-185

Preprint. Discussion started: 21 October 2025

© Author(s) 2025. CC BY 4.0 License.



WIND
ENERGY
SCIENCE
DISCUSSIONS

Moreover, an unintended benefit of yaw misalignment of the first turbine is the secondary wake steering effects (King et al., 2021), where its wake can deflect the wake of the downstream aligned turbine, potentially increasing the power output in wind farms (Fleming et al., 2018). Since secondary wake steering is strongly dependent on the spanwise velocity in the wake—and background wind veer can substantially alter this velocity component—these findings have direct implications for the design and optimization of wake steering strategies in wind farms.

Code availability. Codes used to generate the plots will be made available upon request if the paper is accepted for publication.

Data availability. All data related to the cases analyzed will be made publicly available through 4TU Research Data (upon acceptance of the paper) at https://doi.org/10.4121/e38ae0af-860a-46f0-85af-f384f3cd7d34 (Purohit et al., 2025).

Author contributions. SP developed the methodology, carried out the experiments, performed data analysis, and wrote the paper. HS helped the first author in carrying out the experiments. AS shared his expertise in setting up the SPIV setup, provided scientific supervision throughout the analysis phase, and revised the manuscript. WY contributed towards funding acquisition, scientific supervision throughout the analysis phase, and revising the manuscript.

Competing interests. The authors declare that they have no known competing interests.

Acknowledgements. The authors would like to thank the amazing technicians in the Aerodynamics lab — Frits Donker Duyvis, Peter Duyndam, Stefan Bernardy, and Dennis Bruikman — without whom the experiments would not have been possible. The authors would also like to acknowledge the help provided by David Bensason, Brian D'Souza, Clem Li, and Kiran Sripathy in setting up the experiment.

*Financial support.* This research is supported by the DIAMOND (DynamIc yAw Models fOr wiND turbine/farm design) project funded by the Dutch Research Council (NWO), Netherlands, through Open Technology Program (OTP) under Grant Agreement No. 20052.

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
