# Peer review of "Wind tunnel study of yawed porous discs subjected to veered inflow"

_Wind Energy Science, 2025_

## Referee Comment (RC3)

**General Comments**

The article discusses the effects of wind veer on wind turbine wakes and available power. The manuscript is well-written and the presented results have valuable potential within both the wind energy community and fundamental wake research. However, some comments should be addressed prior to publication as provided below:

**Specific Comments**

1. Lines 19-29: Veering is stated to be "the change in wind direction with height" and an example maximum of rotor diameter is mentioned. However, the extent to which height-relative veering would impact operational turbines is not clear. For example:

   (a) In Line 25, the statement "At such heights, wind veer can be substantial..." does not follow a description of turbine heights, only mention of the max rotor diameter of 260 m.
   Specific/quantified examples of typical turbine heights from ground (ie. hub height, rotor swept extents) and height-relative veering would clarify physical scale and strengthen motivation.

   (b) Height-relative turning $D(z)$ is reported in the Van Ulden and Holtslag (1985) study as the angle of turning compared to the height $z = 20$m from the ground. Based on their Table 2 and Equation 51, it appears that turning angle variation per-meter (*i.e.* $\partial(D(z))/\partial z$) are greater at lower elevations and begin to asymptote at greater heights. Thus, one may argue that veering across just the rotor swept extent for an operational turbine may be more minimal at greater heights. Alongside clarification of turbine physical extents for Comment 1(a), discussion of possible ranges in typical wind-turning estimates across the rotor of a typical turbine may also strengthen motivation.

2. Line 33: "Large-eddy simulation (LES) is the most preferred way to study wind turbine wakes due to its ability to resolve large-scale turbulent structures and accurately capture unsteady wake dynamics (Xie and Archer, 2017)." This should be restated to remove subjectivity and to be more precise. Specifically, one could argue that LES is not always the most preferred way to study wind turbine wakes when small-scale features desired.

3. Lines 103-112: Expanded discussion on porous disks is needed to motivate why/how they are used as turbine analogs, as well as the effects/consequences of porosity and void location/distribution on turbine wake-mimicking behavior.

4. Section 2.2: Porous disc and wind veer model

   (a) Further description and/or an annotated diagram of the disk design is suggested beyond references to diameter and porosity ratio. Including information on porosity void distribution and void dimensions throughout the disk is advised since this is known to affect wake features. It is also suggested to include the disk position distance in $x$ from the tunnel exit.

   (b) Discussion of the PIV plane coordinate system should be revisited/edited as it appears to be somewhere misrepresented. For example, the diagram in Fig. 1b suggests that the streamwise planes are situated in $xz$, but reported plane orientation in the caption and text is $xy$.

   (c) Visual clarification on orientation of veering blades would be helpful. For example, is the veering blade length in $y$?

5. Section 2.4 Flow measurement system:
   What convergence analysis was done to ensure that 100 snapshots was sufficient? While not impossible to achieve at N=100, many more images are typically needed for PIV data to fully converge – especially for terms involving velocity fluctuations (*e.g.* momentum, TKE). Uncertainty analysis through $\sigma_u$ (Section 2.5) can be an unreliable measure of PIV data "goodness" in the case of non-converged fluctuations.

6. Section 2.6 Flow characterization

   (a) Line 232: "A minor spanwise velocity component of approximately 2 % is observed, likely caused by a slight misalignment of the laser sheet with the wind tunnel exit." This comment leads to two questions

i. If this was due to a laser sheet misalignment, why would the profile exhibit a central hump of higher magnitudes bounded by $v/U_{ref} = 0$ rather than linear asymmetry across the profile? Specifically, above $z/D = 0.5$ and under the legend (only slightly visible – see Technical Corrections 3a), it appears that $v/U_{ref} = 0$.

ii. Could non-zero spanwise and/or vertical velocities in the "clean" case be a result of the open jet-type wind tunnel? Was the $w$ velocity component also plotted for reference?

(b) It may be helpful to include in-text how veer was calculated for Fig. 4.d.

(c) Line 241: "Despite this, the turbulence intensity across the disc area remains below 2 % at all measured streamwise locations for both veered cases...". This does not appear to be entirely true for the 10° veering case at $x/D = 7$.

**Technical Corrections**

1. In-text citations should be corrected throughout the manuscript to ensure parenthetical citations are not in direct reference form. For example, in Line 68: "The aerodynamics of yawed wind turbines were first studied experimentally by (Grant et al., 1997) and (Grant and Parkin, 2000)."

2. Velocity variables in axis labels and text should be edited to denote averaging where appropriate and remain consistent throughout. For example, contours and Fig. 4 profiles show velocity as lower-case $u$ with no over-bar, but streamwise velocity is also referred in text as $U_x$ and $\overline{u}$.

3. It is suggested to eliminate phrases such as "As expected...", "As anticipated..." from the manuscript without citing the source(s) of the expected outcome. If results mimic behavior seen in other studies, it is best to cite those studies rather than assume reader knowledge.

4. Corrections to Fig 4:

   (a) Legends should not block profiles.
   As an additional aesthetic suggestion: consider consistent legend positioning in subplots and/or minimize repetitive legends when colors/line-style represent the same quantities throughout.

   (b) Axis ranges for $u/U_{ref}$ should be corrected to make all reported data visible. Also, minimizing unused high-range space would more clearly show variations between "clean" and veering cases.

5. Lines 197-199 are redundant as the phrase "It should be noted..." is used in two consecutive sentences.

6. Line 265-268: "As the tower is relatively thicker than the diameter of the porous disc (with a disc-to-tower diameter ratio of 10)...". This statement may intend to read differently, but in its current form is self-contradictory and unclear.

7. Line 322: It would be helpful to follow the statement of "...as reported in multiple field experiments..." with direct citation(s) where this is reported.

8. Line 328 "As also discussed in Section 3.1,...": No reference to vortex pairs was made in Section 3.1.